# “We Just Take Care of Each Other”: Navigating ‘Chosen Family’ in the Context of Health, Illness, and the Mutual Provision of Care amongst Queer and Transgender Young Adults

**DOI:** 10.3390/ijerph17197346

**Published:** 2020-10-08

**Authors:** Nina Jackson Levin, Shanna K. Kattari, Emily K. Piellusch, Erica Watson

**Affiliations:** 1Department of Anthropology, School of Social Work, University of Michigan, Ann Arbor, MI 48109, USA; 2Department of Women’s and Gender Studies, School of Social Work, University of Michigan, Ann Arbor, MI 48109, USA; skattari@umich.edu; 3School of Social Work, University of Michigan, Ann Arbor, MI 48109, USA; emkaypie@umich.edu (E.K.P.); nagye@umich.edu (E.W.)

**Keywords:** chosen family, queer family, queer health, transgender health, care, mutual aid, Q/T

## Abstract

“Chosen family”—families formed outside of biological or legal (bio-legal) bonds—is a signature of the queer experience. Therefore, we address the stakes of “chosen family” for queer and transgender (Q/T) young adults in terms of health, illness and the mutual provision of care. “Chosen family” is a refuge specifically generated by and for the queer experience, so we draw upon anthropological theory to explore questions of queer kinship in terms of care. We employ a phenomenological approach to semi-structured interviews (*n* = 11), open coding, and thematic analysis of transcriptions to meet our aims: (1) Develop an understanding of the beliefs and values that form the definition of “chosen family” for Q/T young adults; and (2) Understand the ways in which “chosen family” functions in terms of care for health and illness. Several themes emerged, allowing us to better understand the experiences of this population in navigating the concept of “chosen family” within and beyond health care settings. Emergent themes include: (1) navigating medical systems; (2) leaning on each other; and (3) mutual aid. These findings are explored, as are the implications of findings for how health care professionals can better engage Q/T individuals and their support networks.

## 1. Introduction

“Chosen family” is a term employed within queer and transgender (Q/T) communities to describe family groups constructed by choice rather than by biological or legal (bio-legal) ties. Chosen family implies an alternative formulation that subverts, rejects, or overrides bio-legal classifications assumed to be definitive within an American paradigm of kinship [1]. The provenance of the term “chosen family” in social science discourse derives from anthropologist Kath Weston’s *Families We Choose: Lesbians, Gays, Kinship* [2]. In this watershed project, Weston describes the central role that close friends played in the lives of sexual minorities who often experienced distance or rejection from their families of origin [3]. Using ethnographic and autoethnographic methodologies, Weston takes up kinship as the lens through which to understand “*how lesbians and gay men experience otherness and negotiate their relationship to the outside world*” ([4], p. 976). In her efforts to investigate what “family” means and has meant to lesbians and gay men in the United States, Weston deliberately treats family “*not so much as an institution, but as a contested concept, implicated in relations of power that permeate societies*” ([4], p. 3). Weston’s argument about families of choice is undergirded by a contentious relationship with Schneider’s classic anthropological study on “American kinship” [1]. Schneider proposed that symbolics grounded in the division between the order of nature (i.e., shared biogenetic substances) and the order of law (i.e., code for conduct) define family relations in a United States (US) based context. Queer relationships—which are neither grounded in biology nor procreation, and often operate outside the legal domain—cut across these categories, complicating Schneider’s fundamental claim that family ties are reckoned between poles of blood and law. As such, queering kinship obligates critical engagement with “the family’s” traditionally ascribed organization and authority across the social landscape ([5], p. 3).

Weston’s empirical on work on families of choice takes place against the backdrop of the HIV/AIDS crisis in San Francisco in the late 1980s. The ethnographic context of her findings is linked inextricably to her theoretical contributions. That families of choice and an ethics of care in the context of health and illness intersect both practically and theoretically is not incidental. In fact, Weston’s investigation was initially inspired by the question of mutual aid during and after the emergence of the HIV/AIDS epidemic [4]. When families of origin rejected responsibility for end-of-life care of terminally ill people living with HIV (PLWH), gay and lesbian patients were cared for primarily by friends [6]. Such an act of custodianship elevated the relationship from friendship to something more; an iteration of family. For the purpose of this study, illness and care in the lives of Q/T individuals moves beyond the context HIV/AIDS. Nevertheless, the fact that the analytic term “chosen family” emerged in tandem with the act of providing care at the threshold of life and death theoretically sutures the practice of “choosing family” to the notion of care writ large. By linking the study of care [7] to questions of new formulations of family, the investigation presented here offers critical implications for research and practice in the fields of health and human services. It is important to build a research and practice vocabulary that expresses, interprets, and operationalizes the relationship between *care* and *family* for a contemporary context, particularly for families that are constructed beyond bio-legal ties. 

Weston’s seminal inquiry into families of choice in 1991 serves as a canonical reference for many subsequent works that take up the question of “chosen family” within and beyond the US context, including this one. Studies drawing upon Weston’s work have explored queer kinship through multidisciplinary lenses [8] and pluralistic contexts, including domestic and interpersonal labor within Q/T partnerships [9] and communities [10]; in pedagogical relations [11]; and within queer friendships [12]. Intersectional understandings of chosen family also characterize this topic of study. Representations of Black queer kinship in ballroom culture have ascended in popular media with the advent of the television phenomenon *Pose* [13,14], and have been explored empirically through ethnographic inquiry [15]. A 2019 photography exhibition at the GLBT Historical Society titled “Chosen *Familia*” engaged the public sector in a visual interrogation of chosen family by reinterpreting the traditional family photo album in order to reframe and document Latinx queer families [16]. Notions of kinship and care are often informed by intersectional mutual aid practices that proliferate amongst marginalized communities, such as immigrant communities [17], poor communities, and/or communities of color [18,19]. Mutual aid, discussed further in our results, directly engages the social conditions a community seeks to address by providing housing, food, health care, or transportation in a way that draws attention to the politics creating need and vulnerability [20]. Nicolazzo et al. ([21], p. 307) note that queer individuals who negotiate kinship networks under varying degrees of adversity often form *“counterhegemonic cultures of care.”* In these informal networks, care and support flows freely without the regulation of rigid legal, biological, or social ties. Such practices are evidenced in many different types of marginalized communities [22]. Therefore, while “chosen family” is not an exclusively queer experience, the phenomenon of care giving outside the bio-legal family framework has been shown to run parallel along multiple, intersecting lines of social disenfranchisement.

In the case of queer chosen family, relations of choice and relations of origin are complicated by the fact that, as Weston identified, chosen family often *complements* rather than *replaces* bio-legal family [3] (emphasis ours). The notion that all queer people are cast out upon coming out perpetuates a belief that being queer or transgender warrants repulsion; that coming out is primarily met with disapproval or rejection from one’s family of origin [23]. Rather, many Q/T folks relate with chosen families and with their families of origin, while some relate exclusively with chosen family as a primary kin network. Furthermore, legal rulings such as *Obergefell v. Hodges* [24] have altered the backdrop against which Q/T people negotiate relationships with bio-legal family and family of choice. Such a decision has implicated the ways in which same-sex partners define and operationalize chosen family in terms of intimate partnership, given the changing—and contingent—nature of access to the legal construction of family for Q/T people.

For contemporary social work, health care and other human service professions, the domain of knowledge surrounding queer chosen family and care giving in terms of health and illness is lacking and requires further investigation. At stake for families of choice are the ways in which social structures governing access to resources “*interpolate the symbolic oppositions which inform queer kinship into everyday experience*” ([2], p. 5). In the US, lesbian, gay, bisexual, queer, transgender and asexual individuals can be legally denied housing in 24 states based on sexual orientation and in 23 states based on gender identity; denied employment in 22 states based on sexual orientation and 23 states based on gender identity; denied public accommodations in 26 states based on sexual orientation and 25 states based on gender identity [25,26]. These challenges are also evident in health care settings, with only 37% of lesbian, gay, bisexual and transgender individuals living in states that guarantee health insurance protections regarding gender identity and sexual orientation [27]. Known impediments preclude this population from accessing basic health care, let alone from accessing healthcare that provides coverage for health needs related to sexual orientation and/or gender identity.

Challenges for individuals in this population are not limited to policies. Q/T individuals report encounters with practitioners who prohibit their partners from joining them at office visits, avoid eye contact, make assumptions about their sexuality and gender, put their bodies on display without consent, and a bevy of other horrifying concerns [28]. In 2015, a third of transgender and nonbinary individuals in the US report negative interactions with healthcare providers based on their transgender identities within the past year [29]. Queer individuals often struggle with locating and accessing a queer affirming provider, and even when they do, may face barriers to bringing their whole authentic selves into healthcare settings [30]. 

Because access to appropriate and affirmative medical care is empirically challenging for this population, we intend to explore notions of care giving amongst Q/T individuals that supersedes formal medical contexts. Our goal is not to investigate informal medical care; that is, practices of performing medical procedures or circulating medical counsel outside of formal medical systems. Rather, we aim to better understand the expansive notion of care giving and receiving amongst Q/T individuals and their chosen family members, and the ways in which serving as stewards of one another’s health and well-being may be constitutive of chosen family formations. Therefore, the goal of this exploratory phenomenological study is to examine the concept of “chosen family” amongst queer and transgender identifying adults. Our specific aims are to: (1) develop an understanding of the beliefs and values that form the definition of “chosen family” for queer and transgender individuals over the age of eighteen; and (2) understand the ways in which “chosen family” functions in terms of care regarding health and illness. 

It is noteworthy that in our recruitment methodology, discussed below, we did not explicitly indicate inclusion criteria related to having disability(ies) and/or illnesses; nor did we overtly advertise our investigative attention to the concept of care within the phenomenon of chosen family. Nevertheless, (*n* = 10, or 90%) of participants identified as having one or more physical or mental health disability(ies) and/or illnesses. This unexpected demographic quality of our sample greatly informed our findings about participants’ basis for constituting chosen family structures as well as the ongoing practices of care giving and receiving that define them. Theoretical alignment between the element of *“otherness”* implicated in both queerness and disability makes for a *“complementary relationship”* between these identities ([31], p. 139). Therefore, in this study we work to build empirical evidence that substantiates a dynamic relationship between informal care, family, queerness, and disability(ies) and/or illnesses. Such evidence may inform and improve upon formal services provided to this population by health and human service professionals.

## 2. Materials and Methods 

In alignment with our commitments to critical methodology, which calls for continuous examination of the assumptions that undergird research resources and inform inquiries and findings [32], critical attention to author identities and positionalities characterized the research process throughout. Therefore, in articulating our materials and methods, we begin by offering reflexivity on our researcher positionalities: The study is led by a doctoral candidate who identifies as a cisgender, secular, fluid, well-resourced, able-bodied, white woman, and a multidisciplinary scholar. The study is guided by a faculty mentor who identifies as a critical intersectional feminist, mixed methods scholar, and a queer, white, cisgender, Jewish, middle class, disabled, chronically ill, neurodivergent, femme. The study is supported by two Master’s level student investigators, one who identifies as a white, working class, disabled, neurodivergent, queer, bisexual woman; one who identifies as a bisexual, polyamorous, white, currently able-bodied woman.

The inception of this study began as a summative project for a doctoral seminar addressing social work policy and practice for marginalized populations. Excitement about the project led the student and faculty authors to pursue collaborative investigation following the completion of the seminar, which included applying for and receiving funding from the campus queer center. Conducted in 2019 in two adjacent counties in a Midwest state, this study was approved by Institutional Review Board (IRB) at a local public research university. We used a modified phenomenological approach [33,34] to explore the lived experiences of Q/T adults (eighteen-years-old and above) who identify as participating in “chosen family.” The sampling method included posting a call for participants in multiple queer social media groups on Facebook, Twitter, and Instagram, and on email to academic listservs, in pursuit of a sample of ten to fifteen participants. This call for participants articulated inclusion criteria as: adults over the age of eighteen who self-identify as lesbian, gay, transgender, queer, intersex, asexual, two-spirit, or another non-heteronormative sexual or gender identity, (LGBTQIA2S+) and who participate in chosen family; and live in one of the two adjacent counties where the study took place. Flyers were circulated in both images and plain text formats for accessibility.

The student authors conducted loosely structured interviews (*n* = 11). The interview protocol employed contained questions to examine participants’ experiences of “chosen family.” Interviewers used follow-up probes based on each participant’s response. Participants completed a demographic screening sheet containing questions regarding their pronouns, age, race/ethnicity, gender identity, sexual orientation, income level, highest level of education, and disability(ties) and/or illnesses. Interviewers collected these attributes for two reasons: firstly, to ensure correct use of identities in writing up findings; and secondly, to potentially track trends that may appear among or across certain identities [33]. Interviews took place in-person in participant’s homes, local public spaces, in a private room at the university, or on confidential video conference software, per participants’ preference. Interviews were recorded following completion of signing informed consent forms as well as providing on-record verbal acceptance of being recorded. Each audio file was transcribed verbatim by an author separate from the person who conducted the interview, and interviewers reviewed transcripts for accuracy.

Analysis began with the doctoral candidate using inductive coding by hand for three of the interviews, thereby creating an initial code list. Next, the faculty mentor performed consensus coding via second cycle coding on an un-coded transcript [35]. The doctoral candidate and faculty mentor met to deliberate codes. All discrepancies were discussed, and consensus was reached on all accounts. The doctoral candidate and Master’s students used a refined code list to code the remaining interviews. Once all interviews were coded, the doctoral candidate and faculty mentor used a digital tabletop method of individual theme development to identify themes and patterns [35]. This digital version of the tabletop method is a COVID-19 informed adaptation that involves screen sharing between researchers, using virtual color-coded “sticky notes” which were systematically re-ordered to achieve thematic saturation. Finally, consensus between the doctoral candidate and faculty mentor around the themes was reached and resulted in the themes discussed below. 

The sample is comprised of (*n* = 11) individuals (see Table 1 for sample demographics). 

Through a combination of select-all-that-apply and self-report options for gender identity and sexual orientation, participants reported gender identities that include Woman (36.30%); Woman AND Nonbinary (9.10%); Transmasculine (9.10%); Nonbinary (18.20%); Transmasculine AND Nonbinary (9.10%); Transmasculine And Nonbinary AND Genderqueer (9.10%); Transmasculine AND Agender (9.10%). Participants’ sexual orientation/identity included Bisexual (45.50%); Pansexual (9.10%); Bisexual AND Queer (9.10%); Bisexual AND Queer AND Pansexual (9.10%); Pansexual AND Asexual (9.10%); Pansexual AND Demisexual (9.10%); Pansexual AND Queer (9.10%). While we recruited for participants who identify as LGBTQIA2S+, our sample is comprised of participants who identify as queer and trans (Q/T). Therefore, in this paper we use the acronym Q/T to accurately reflect our sample. We acknowledge that queer is a capacious term with philosophical, political, practice-based, and identity-based implications. Our use of “queer” in this paper is an in vivo term derived from participants’ self-reported sexual orientation/identities, referring to non-cisheteronormative gender and sexual identities and practices. 

Participants range in age between 18–25 (27.30%); 26–35 (63.60); 36–45 (9.10%). Using select-all-that-apply options, the majority of participants identify as White (54.50%), and several participants identify as another race or ethnicity AND White (Asian and/or Pacific Islander AND White, 9.10%; Middle Eastern, North African, and/or Chaldean AND White, 9.10%; American Indian, Native Alaskan and/or Native Hawaiian AND White, 9.10%). Asian and/or Pacific Islander, Filipino, Chinese identifying participants comprised 18.20% of the sample. Participants are highly educated (Doctorate degree or Professional degree 27.30%; Master’s degree 36.60%; Bachelor’s degree 18.20%; High School diploma 9.10%; General Educational Development 9.10%). Nevertheless, participants also reported low levels of annual income (45.50% earn below $20,000; 18.20% earn $20,001–30,000; 27.30% earn $40,001–$60,000; and 9.10% earn $60,001–$80,000). While level of education often serves as an indicator of socioeconomic status in the general population [36,37], this proxy is inappropriate for Q/T populations who face barriers to employment and wealth attainment which are not necessarily moderated by level of education [38]. Additionally, the inflated education level is attributable to the geographic context in which the study took place (a small town primarily organized around a research university). 

In terms of disability(ies) and/or illnesses, participants self-reported a range of identities. As noted in the introduction, all but one participant in the sample indicated experiencing disability(ies) and/or illnesses: Chronic pain, mental health, hard of hearing (9.10%); Depression/anxiety, post-traumatic stress disorder (9.10%); Mental health disability (9.10%); Post-traumatic stress disorder, generalized anxiety disorder, premenstrual dysphoria disorder, depression (9.10%); Severe mental illness, physical impairments (9.10); Autism spectrum, chronic illness, lupus (9.10%); Depression (9.10%); Epilepsy, autism, chronic anxiety (9.10%); Mad/neurodiverse (9.10%); Obsessive compulsive disorder, anxiety, depression (9.10%); N/A (9.10%). In recognition of the fact that diagnostic schemas have been historically utilized as oppressive instruments [39,40], we intentionally allowed participants to share the disability(ies) and/or illnesses with which they identify, as opposed to framing their health status in terms of conditions for which they have received medical diagnoses. This decision is especially important in the case of neurodivergence, which—rather than a diagnostic category—is a social movement that venerates the paradigm that there is no one “right” or “normal” way to be human [41,42].

Finally, it merits stating that that our sample size (*n* = 11) is appropriate for our methodology (exploratory phenomenological inquiry). For readers who are oriented toward quantitative methodologies that seek generalizability and therefore require large sample sizes, our sample may appear to lack rigor. However, exploratory qualitative methodologies that seek saturation over generalizability are best carried out with small sample sizes in order to maximize depth of understanding of the topic under study [43]. In our case, the topic of chosen family is best understood through the in-depth responses of our informants, to whom we refer throughout by their pseudonyms (in order of appearance): Minnie, Grey, Andres, Clive, Lydia, Chrissy, Tish, Chase, Elle, Jasper, and Maria. The following sections present findings of our informants’ experiences of participating in chosen family.

## 3. Results

Six major themes emerged from the data: (1) navigating medical systems; (2) leaning on each other; (3) mutual aid; (4) chosen family enactment; (5) chosen family embodiment; and (6) key components of chosen family. The first three relate to health and care giving, and therefore will be discussed in this article. The latter three relate to broader conceptual dimensions of chosen family as a sociocultural object and will be discussed at length in a separate manuscript. 

### 3.1. Navigating Medical Systems

Navigating medical systems emerged as a major theme in response to interview questions about the ways in which participants experience support for and from their chosen family members when engaging formal health care services, providers, and institutions. Amongst the topics that characterize the theme of navigating medical systems, the most salient issues include experiences of medical trauma and decision-making regarding emergency contacts.

#### 3.1.1. Medical Trauma

Several participants raised accounts of medical trauma which prompted them to either divest from formal medical care altogether or cautiously ensure that at least one chosen family member (i.e., an advocate) would accompany them to future medical appointments. One participant, Minnie, a bisexual cisgender woman who identifies as mad and neurodiverse described negative experiences with medical institutions that prompted her to divest from formal medical care: 

*I have avoided medical appointments because I have a lot of institutional trauma and experiences of institutional harm… to be honest, the mental health system is demonstrably an oppressive piece of shit* [sic]*… I received no value from that particular diagnostic schema, I challenge it—a lot. I don’t use that.*

As a cisgender woman, Minnie’s medical trauma was more explicitly linked to her neurodiversity (what she refers to as “mad pride”) rather than her sexual or gender identity. However, several participants, particularly those who are transgender and/or nonbinary described medical trauma that directly implicated their sexual and gender identity. Grey, a transmasculine person who identifies as pansexual and asexual, and who also identifies with experiencing depression, discussed planning his upcoming top surgery with his chosen family. Such plans involve coordinating efforts to ensure that at least one chosen family member—an advocate—would accompany his hospital visits:


*When I have top surgery, I will have to have somebody in the hospital with me pretty much at all times. Mostly just because people at [the local research hospital] aren’t necessarily the greatest about gendered care, aside from the people who work in the gender services department. And so, I’ve been told that it’s important that I have a person there, for—at least to be my advocate while I’m not feeling the greatest.*


Like Minnie and Grey, Andres experienced medical trauma prompted by both his sexual and gender identity and his neurodiversity. Andres is a transmasculine and nonbinary person, who identifies as pansexual and demisexual, and also identifies with being on the autism spectrum and having chronic illnesses including lupus. Describing a recent lupus flare up, Andres said he “really hates” going to the hospital because “as somebody on the spectrum [he doesn’t] do very well in hospital settings.” In elaborating further, Andres described a graphic event that involved restraints, unnecessarily:

[Exasperated sigh]. *Doctors can have…a hard time explaining what they’re doing before they do it, before they touch you. It can be very sudden and very bright and very loud and just overstimulating, overwhelming and…as somebody whose transgender I’ve had really negative experiences in the hospital in the past, that, um*—[they] *really didn’t understand my gender identity,* [they] *actually put me in a private room, handcuffed to a bed because they didn’t know where to put me, when I was around 19… so I just really don’t like hospitals. I try not to go.*

Following this traumatic experience, Andres noted that his chosen family member, Ray, attends every doctor’s appointment with him to advocate for him in medical settings. One such instance of Ray advocating for Andres included the occasion when a provider insisted upon taking Andres off his hormone replacement therapy (HRT) “*for reasons that didn’t make any logical sense*.” Ray solicited support from the hospital social worker who brokered with physicians in order to allow Andres to continue his HRT. Despite Ray’s deft and proactive advocacy, Andres felt that doctors take the accompaniment of his chosen family members “*less seriously*” than they would a bio-legal family member advocate.

Clive, a transmasculine and agender person who identifies as bisexual, and who identifies with having severe mental illness and physical impairments, described needing a chosen family member advocate to accompany him to his upcoming top surgery. In describing the chosen family member who he would bring to his top surgery, Clive considered one of his fellow transmasculine friends who had already vetted several doctors. About his advocate, Clive said: 

*He did the research,* [and has] *been through those kind of shitty experiences before me where it’s like—yes, they will do top surgery but they won’t respect your pronouns the whole time and they might fudge it up a little bit because they’re used to doing breast reductions* [in contrast to top surgeries as part of medical transition].

Clive’s advocate found a doctor in a nearby city who offers both affordable and adequately trans-friendly services, and Clive described his approach to bringing his advocate as a companion for his surgery: 

*I think that would actually be a really good idea* [to bring an advocate]. *I was actually going to do it by myself because usually I’m pretty uncomfortable with people just in general knowing any part of my medical stuff, but I trust him. And especially with—I have a hard time advocating for myself in* [medical settings]*, so I think having someone there who would consistently correct pronouns, or—you know, someone who has no skin in the game…Because, for me, if I speak up they’re gonna not want to help me or whatever… There are doctors that are like that. Once they find out that* [top surgery] *is what it’s for, they won’t do it.*

Like Clive, Lydia and their partner tapped into a whisper network to find a “*trans-friendly*” physician who would provide appropriate care. Lydia identifies as a woman and nonbinary, as bisexual, pansexual, and queer, and as experiencing obsessive compulsive disorder (OCD), anxiety, and depression. They described attending back-to-back gynecological exams with their partner. Because Lydia’s partner is also gender variant, they “*wanted someone who is trans friendly.”* The pair found a gynecologist who made them feel “*validated*” in the way they were able to access care without having to “*explain much*” about their relationship in the appointment:

[The provider] *was just like, ‘Oh, you were with this person in the last appointment. Cool.’ He was also very open, asking if I had other partners and things like that. I identify as monogamous, but it was really cool that he even knew to ask.*

While several participants in the sample raised accounts of acute medical trauma, Lydia’s testimony served as the sole description of a positive experience of interacting with a medical provider. However, their interview also included an account of fearing that their partner would be discriminated against when accompanying them to urgent care due to an isolated case of food poisoning during a brief period when they lived in a conservative region of the country.

Participants experienced medical trauma prompted by their mental health status, and sexual or gender identity. Coping mechanisms for mitigating future traumatic encounters when seeking formal care included divesting from institutionalized medical care; bringing a chosen family member with them to appointments and procedures to serve as an advocate; or soliciting insider knowledge from a whisper network of queer chosen family and community members who pre-vetted various providers. 

#### 3.1.2. Emergency Contacts

Our interview protocol probed participants about who they keep as an emergency contact, who keeps them as an emergency contact, and how they go about the process of making and communicating this decision with their chosen family members. Responses to this question varied, with some participants including partners, some including bio-legal family members, and some including chosen family members. Across participants, the extent to which emergency contacts are mutually kept (i.e., two people serve as each other’s emergency contacts) varied as well. This trend prompted us to develop codes for symmetry and asymmetry within chosen family relationships. Symmetry refers to mutually held, shared responsibilities and an even distribution of care efforts across two or more people; asymmetry denotes the opposite, in which one person carries more responsibility than the other with respect to a particular domain of the relationship.

In many cases, participants held symmetrical emergency contact relationships with their partners. Chrissy is a bisexual and queer cisgender woman who does not identify with any disability(ies) and/or illnesses. She described her chosen family as her “*framily*” (a combination of the words “friend” and “family”). Chrissy’s *framily* is comprised of six women who also identify as lesbian or bisexual. The *framily* formed over the course of fifteen years, starting at a time when they were all living in the same West Coast city. At various points over the years, each *framily* member held at least one of the others as an emergency contact, depending upon who was partnered, single, or living together as roommates at various time points. Now, Chrissy and her wife, Shae, list each other as symmetrical emergency contacts. However, one of the six *framily* members experienced the onset of a chronic illness which jeopardized her employment. As Shae specializes in employment discrimination, Chrissy’s *framily* member held Shae as an asymmetrical emergency contact. Chrissy’s case illustrates the ways in which various symmetrical and asymmetrical emergency contact relationships may exist within a chosen family group.

Similar to Chrissy, Tish also holds a symmetrical emergency contact relationship with their current partner. Tish is a nonbinary person who identifies as pansexual, and who identifies with experiencing post-traumatic stress disorder (PTSD), generalized anxiety disorder (GAD), premenstrual dysphoric disorder (PMDD), and depression. Tish maintains no extant relationship with their family of origin and before beginning their current partnership they frequently experienced a dearth of relationships to hold as either symmetrical or asymmetrical emergency contacts:

*There’s a lot of forms that I filled out in the past* [where I didn’t] *have an emergency contact and someone’s like, ‘You don’t have anyone?’ and I’m like, ‘No, I don’t. Can I put your name?’* [Laughs]*. Like, I’ve literally asked that of managers before. They’re like, ‘Yeah, I guess, like, I’m not gonna say no.’ But, I do finally have someone that I can put as my emergency contact—my partner, I’m pretty sure I’m his, too.*

Similarly, Grey’s relationship with his family of origin is tenuous. On one occasion when he experienced a health emergency and needed urgent clinical care, he realized that his mother was his default emergency contact despite their estranged relationship:


*That was a time in my life that I wasn’t talking to my mother and I realized it was kind of absurd to still be putting her as my emergency contact when I wasn’t actually talking with her, cause I was like ‘golly that would be an uncomfortable phone conversation or an uncomfortable interaction,’ if something did happen to me and all of a sudden my mother shows up.*


In the waiting room of the urgent care clinic, Grey then texted his chosen family member:


*I was like ‘Hey, this is a really weird question but, is it ok if I put you as my emergency contact? Cause you’re the person I think of when—if I’m sick and something goes wrong, you’re the one I would call. Is that ok?’ And she was like, ‘Yeah, absolutely, that’s totally fine!’ And—now she’s my emergency contact.*


Grey now holds an asymmetrical relationship with his emergency contact, who holds a symmetrical relationship with her fiancé. Despite this, Grey attests that—in truth—when things go “*not great*,” he is the one she is most likely to call upon, rather than her fiancé.

Clive also experiences an estranged relationship with his family of origin. Nevertheless, he holds an asymmetrical emergency contact relationship with his mother. About his mother, Clive stated that she’s “*definitely the one person I would trust in an emergency, just because she knows my entire history*.” He was especially adamant about his emergency contact being deeply knowledgeable about his medical history because his legal name and gender are marked differently in his medical records than to the name and gender with which he currently identifies. “*If for some reason I was incapacitated in a hospital,* [my mom] *would have my records…I wouldn’t have the right name on the forms*.” Furthermore, in spite of their estrangement, Clive trusts that his mom “*would come at the drop of a hat, whereas my friends are all very busy and might not be there*.” Of all of the participants, Clive was newest to the experience of chosen family, and new to the particular group of queer friends with whom he was developing family bonds at the time of the interview. As a result, he felt safest holding his mom as his emergency contact, despite their otherwise distant relationship.

Similarly, Chase holds their mom in an asymmetrical emergency contact relationship because for Chase, their mom is the one who knows all their medical records. Chase is a transmasculine, nonbinary, and genderqueer person who identifies as pansexual and queer, and who identifies with having a mental health disability. Chase maintains an amicable relationship with their family of origin. While they rely upon their parents for contacts in the event of medical and financial emergencies (Chase described an instance when they were robbed and called their dad to provide logistical support in recovering financial information), they explicitly distinguish between an emergency contact and a *“crisis contact.”* For Chase, a crisis contact is someone to call upon for emotional support:

*I think* [crisis contact] *is different than emergency contact. When I think of emergency contact, I’m thinking of, like, ‘I’m bleeding on ground!’ But I think crisis contact is something different. It’s like, ‘Oh my god, I’m having a horrible day.’ My* [crisis contact] *is going to be the one that I call. And, vice versa.*

Elle distinguished between an emergency contact and a crisis contact as well. Elle is a bisexual cisgender woman who identifies with having depression, anxiety, and PTSD. She and her two roommates share a round-robin of emergency contacts between the three of them to achieve symmetry. One of her roommates is her ex-partner with whom she moved across the country for jobs at the same local company. Despite ending their romantic relationship shortly after their relocation, they continue to live together as roommates and chosen family members. Their third roommate’s family of origin also lives in a distant state. Due to proximity, Elle and her roommates arranged for symmetrical emergency contact relationships:

*We talked about* [being each other’s emergency contact]*, because when we were first filling out our* [paperwork] *at work it was like, ‘We literally don’t know anybody else out here.’ What are they gonna do, call his dad whose all the way* [across the country]*? He can’t do anything about it! We’re all from all over the place, so it’s like, yeah to some extent we all kinda do need to rely on each other* [in an emergency].

Nevertheless, Elle also maintains what Chase referred to as crisis contacts. Having established close friendships in the year since her relocation, Elle describes her emergent chosen family members as crisis contacts for *“personal emergencies”;* someone she would call if she were experiencing emotional distress, rather than a medical, financial, or legal emergency.

In one case, a participant discussed taking on a particular legal role for a chosen family member which bore more gravity than an emergency contact: power of attorney (POA). Jasper is a nonbinary person who identifies as bisexual, and who identifies with experiencing chronic pain, mental health, and is hard of hearing. Jasper described the decision to become an asymmetrical POA for their live-in chosen family member as a serious and difficult decision. When first asked to take on this role, Jasper declined. After a year of deepening their chosen family relationship and discussing various dimensions of the decision, Jasper decided to move forward with signing the paperwork: 


*I was pretty nervous but pretty confident. Like, ‘I can vouch for this person. I can do it.’ That was when I was like ‘Oh, someone really considers me to be this responsible, okay…’*


The following year, Jasper moved to a new residence and decided at that time to cease their role as POA. Jasper described shifting out of this role as bringing both relief and sadness, because it marked an official separation from their co-habitation with their chosen family. However, the experience of holding such profound responsibility for someone in their chosen family network was ultimately rewarding:

*We shifted after I moved out because I was like ‘I don’t think I can sustain this if I’m not living with you or in close proximity of you’ … That was also a hard thing too because I felt like that was the last part of me shifting out of the lived-in part of the family, but it was also like, ‘Okay, thank you for letting me experience what that* [level of responsibility] *is like… you’ll always be a part of my life but just not as intensely anymore.’ It was a good space to grow in for sure.*

Emergency contacts amongst chosen family members are various, personal, and contextual. For some participants, emergency contact relationships are symmetrical, while others hold asymmetrical relationships in relation to this concept. Taking on the role of an emergency contact or POA for a chosen family member can be simultaneously rewarding and grave. Some participants prefer their emergency contact to be their chosen family members because of estranged relationships with family of origin, while others opt for family of origin members to serve as emergency contacts for the sake of medical and biographical history (in spite of estranged relationships). Others proactively distinguish between emergency contacts and crisis contacts who provide emotional rather than medical, financial, or legal support in an emergency.

### 3.2. Leaning on Each Other

Leaning on each other emerged as a dominant theme throughout each interview in the sample. Leaning on each other is an in vivo code used to describe informal, reciprocal care giving amongst chosen family members. Discussions of informal care emerged organically and unprompted throughout the interviews, compared with discussions of formal medical care which only emerged in direct response to probes. Providing and receiving care was discussed in the form of organizing around health needs, emotional support, and eating together. In deciphering the vectors of care, we used codes that distinguished between receiving from and giving to chosen family members, as well as providing care for the self.

#### 3.2.1. Organizing around Health Needs

Organizing around health needs was a recurrent topic that emerged throughout the sample. This item represents the ways in which participants’ chosen family members proactively coordinate the provision of care amongst each other generally, and specifically when a member encounters an acute physical or mental health challenge. All participants described using group messaging tools across various online platforms to coordinate their care efforts. Minnie discussed establishing a group message which she titled *“Team Minnie.”* She requested consent from members included in that group to serve as a reliable resource for when she was struggling with her mental health: *“We figured out how to create an easy way for me to reach out because I live alone.”* Several participants who anticipated undergoing top surgery around the time of the interview spoke about the ways in which their chosen families have already developed schedules for feeding, transportation, and bedside aftercare. Grey described this intentional distribution of care across his chosen family network as a *“shared burden of care.”* Lydia discussed *“sharing the burden of care”* for a chosen family member who was grieving the loss of his mother and struggled through his depression to consistently arrive at work on time. Chase discussed the support they received during and after leg surgery as a pivotal experience that laid bare the distinction between friends and chosen family:

*I had friends that created a spreadsheet with all the different things that I needed so I could send it to other chosen family members, so that I could get my needs met. And it was interesting to see who showed up. I think sometimes in those moments it becomes clear where that chosen family is. I do think it’s hard to find what the difference is between chosen family and good friends when you’re out dancing at a club, right? You know, those can look the same. But I don’t think they look the same when* [it comes to…] *dealing with the messy, icky stuff about living sometimes—when people* [are] *willing to step up without even having to necessarily ask.*

Concerted and highly organized efforts to coordinate care among chosen family networks using online and digital communication tools emerged as a prevailing theme throughout the interviews. Care for physical and mental health was the primary focus of “*organizing around health care*” while organizing around emotional and social support fell under the headings discussed below.

#### 3.2.2. Emotional Support

Emotional support emerged from the interviews as the preeminent mode of care that chosen family members give to and receive from one another. In many cases, emotional support directly related to identity-based needs, especially regarding queer and transgender identity. Grey noted that he turns to various members of his chosen family–which is comprised of a non-monogamous *“polycule”*—when needing support about his gender identity:

*If I need emotional support about my gender identity or trans things, then I can sit and talk with my girlfriend* [who is also trans] *for a long time and lean on her for things. And if she’s not available then her wife is genderqueer, and so is their boyfriend’s partner.*

Chase discussed relying on their chosen family for emotional support as opposed to their family of origin, because the shared queer identities between them and their family of choice facilitated unmediated understanding:

*I think that I don’t lean on my* [biological] *family very much for emotional support and part of that is with my chosen family, I don’t I need to go to through all the explaining of queerness, transness, all the other shit* [sic] *that is exhausting to try to define, right? Whereas that’s just already known that we can build off of* [that].

Chrissy discussed emotional support being “*of course mutually given, reciprocally from all* [my chosen family members] *in different ways*,” especially through break ups. Support through breakups was especially important for Chrissy who, like her chosen family members, identifies as bisexual. Having chosen family members who are also bisexual was supportive for Chrissy who found that the liminality of bisexuality often felt isolating even within the Q/T community, and the specific empathy and emotional support of her bisexual chosen family members was significant to their bond. Elle contended that the resource most commonly and significantly shared amongst her chosen family “*comes down to emotional labor*,” especially during acute episodes of depression and anxiety. In addition to queer identity and mental health needs, multiple participants also described providing ongoing emotional support to their chosen family members who had suffered great loss. Elle and Andres each shared an account of supporting chosen family members through the loss of a parent. Maria, a bisexual cisgender woman who identifies as having epilepsy, autism, and chronic anxiety, discussed helping a chosen family member clear out their biological parent’s house following that parent’s death. Jasper described persistently supporting a chosen family member as they suffered through withdrawal from drug use:

*I remember one person* [in our chosen family] *was struggling with withdrawal and we were trying to be there to support, and that was very emotionally taxing but we knew that it was going to be worth it so we just stuck on it anyway. There was a lot of blood, sweat and tears… But we didn’t give up on each other.*

The intersection of mental and physical health, and queer identity was significant in terms of the specific quality of emotional support that participants expressed needing from and providing for their chosen family members. In a poignant statement, Grey exclaimed that, “*I don’t think I know a single queer person that’s not chronically ill in some way*.” Grey’s statement, and the prevailing presence of disability(ies) and/or illness identities in our sample, reinforces the fundamental notion that the type and quality of support at the specific intersection of health and queer identity is one of the essential components for establishing and maintaining chosen family.

#### 3.2.3. Eating Together

Eating together was a pervasive practice raised in every single interview. For many participants, their chosen families formed around shared meals. Tish spoke about their first encounter with “*found family*” when, as a teenager, they spearheaded a Gay Straight Alliance group but was banned by the principal of her conservative high school. As an alternative, they and their queer classmates started having *“little gay picnics”* in the nearby public park. Tish described these picnics as the provenance of their chosen family; as their first and “*most significant instance of queer community really being there for each other*.” Clive noted that following each weekly drag show event at the local queer night club, he and anyone else who stayed out late enough would find themselves together at a diner for a “*family dinner*” in the dawn hours of the morning. Clive described the ritual as gratifying because: “*We’re all feeding each other. We’ve all paid for each other’s meals at some point*.” Chrissy’s chosen family consolidated around weekly Sunday night potluck dinners which rotated between members’ households. Consistent dinners together served as the foundation for establishing family rituals which developed into annual holiday traditions. 

Chrissy described a particular Thanksgiving during which one of her chosen family members’ biological mother was visiting the city and attended the holiday meal. Chrissy noted a distinct discomfort with the juxtaposition between the closeness of her chosen family and the detachment of family of origin:

*It was a little awkward because here we were having this great family dinner, but it wasn’t biological family* [Laughs]*, but it looked like a family dinner and* [the mom] *was the outsider. There was definitely something that I think was jarring to her to see all of these LGBT* [lesbian, gay, bisexual, transgender] *women—these queer women—acting like a family and having that closeness and connection, but her being the only one related by blood and not having that connection at all.*

Elle said that her chosen family came together around making homemade dumplings, a food that is tied to her heritage within her family of origin. The event of dumpling-making feels to Elle like a celebration akin to “*Friendsgiving*,” such that anytime they decide to make dumplings together, shared food becomes the site of enacting and venerating chosen family. Jasper lived with their parents during college but described a nearby house where several of their chosen family members lived as roommates. Weekends frequently occasioned sleepovers which consistently resulted in homemade waffles and pancakes the following morning. Maria noted that cooking for others serves as the ultimate litmus test that distinguishes whether a person is her friend or chosen family member: 

*To me there’s a difference between* [chosen family] *and friends because friends are people that you enjoy being around, but chosen family is more … who comes to my house and eats with me? Who do I cook for, you know? I think ‘who do I cook for’ is a good easy way to tell if somebody’s part of my chosen family or just a friend.*

In each of these cases, gathering around food served as a foundation for establishing consistent, nourishing rituals. While meals were often discussed as practices for celebrating chosen family, feeding each other was also described as a form of mental and physical health care. Minnie described receiving nutritional support from chosen family members when severe bouts of depression prevent her from eating:

*When I’m really quite unwell, I do need people to step in to remind me to eat. Lots of people feed me because I do not feed myself very well when I’m not well. When I’m well I love cooking. But when I’m not well, I definitely don’t eat … so people bringing me food or—just being with me while we cook* [is helpful]. 

Elle echoed the experience of having a partner cook for her when depression becomes so debilitating that she “*can’t get out of bed*.” Tish also discussed the ways in which their mental health significantly improved since forming a chosen family with their partner:

*The fact that every day that I come home from work, he’s cooking dinner—that’s incredible, because like, mental health and making sure you stay fed and that you have a good meal and home cooked meals every night, that is such a huge thing. If he wasn’t around, I’d probably be eating, you know, spaghetti every night—again. My mental health has definitely changed* [as a result].

For some participants, feeding others as opposed to being fed was more salient. Lydia described their practice of ensuring that their chosen family members are well-fed during challenging mental health episodes. In reference to an occasion when they were out to dinner at a restaurant, they ordered food to-go to bring to their chosen family member’s house because they knew he was having a hard time. “*We all kinda take care of each other in that way*,” she said. Andres echoed the sentiment that *“We just take care of each other”* when discussing his routine of bringing food to his chosen family member’s home when he sleeps there on weeknights following late nights at work which preclude a long drive back to his rurally located apartment. The theme of caring for one another through food emerged in both the register of celebration, bonding, and community building as well as in the mode of intervention to depression and other mental health challenges for which providing food was supportive.

### 3.3. Mutual Aid

Mutual aid emerged as a dominant theme throughout the transcripts. As discussed in the introduction, mutual aid has been defined as an ethically charged act of sharing and exchanging material resources. Our use of mutual aid emphasizes the sharing and exchanging of material resources between chosen family members. Material resources encompasses co-habitation, cars and transportation, short-term cost sharing as well as the sharing of structural access to wealth, sharing skills, and sharing time. 

#### Sharing Material Resources

The topic of co-habitation emerged frequently across all interviews. All participants either currently or have previously co-habitated with some permutation of their chosen family members. Minnie discussed the fact that many of her chosen family members have access to generational wealth and own property. Those who own houses share living space with her at no cost. Rather, in exchange she cares for pets, cooks, and cleans, or performs other helpful tasks to buttress the exchange. As Minnie is a naturalized US citizen, she uses one of her chosen family member’s homes as her permanent US mailing address for immigration purposes. Minnie is low income earning and chooses to live a nomadic life. For the majority, she feels safe in doing so and trusts that were she ever to meet her *“worst case scenario”* of experiencing homelessness, someone would offer her a safety net: 

*I still feel insecure sometimes…* [but] *I can probably name between five and ten people who would look after me, who would not see me go onto the street if I was sick or broken, you know? I do know that there would be people who would provide in those sorts of ways.*

Like Minnie, other participants spoke about an open barter system between themselves and those who had access to generational wealth and were able to own property as a result. Clive also spoke about renting space at a low cost from their chosen family member who owns a home. In exchange, Clive takes care of the house and pets. Clive’s chosen family member also helped him purchase a truck from a biological family member at a low cost, and in exchange Clive performs errands as needed. Grey described the long-ranging, collaborative process by which his chosen family is slowly pooling funds oriented toward a down payment on a shared homestead. However, his upcoming top surgery is a more pressing financial priority, and his chosen family is foregrounding that expense before actively pursuing their down payment on a house.

Skills exchanges, for housing or other resources, were discussed in the form of providing specific talents or abilities to chosen family members. Jasper discussed providing regular childcare for his chosen family member who has young children. Childcare included driving the children to and from school, providing after-school supervision, and packing their lunches in the mornings. Lydia, whose profession is in queer student services at a nearby university, described their efforts to create an employment opportunity in their office for their chosen family member who has expertise in sexual assault prevention and education, but who was experiencing temporary unemployment. Chrissy elaborated upon the occasion when her chosen family member experienced an undiagnosed chronic illness, describing in detail the supportive actions her chosen family members took based upon their professional expertise in the legal and medical fields. Tish noted that starting from as soon as they were able to drive, they leveraged their ability to offer transportation to queer community members who needed transportation support for accessing safe spaces: *“If someone needs a ride somewhere, like, yeah sure I’ll take you. Your parents aren’t gonna do it, so I’ll do it.*” Grey narrated a significant event when he helped his girlfriend and her wife unload an oversized generator from their car following an emergency power outage: 

*I ended up going over* [to their house] *and pulling the 200-pound thing into their basement and setting it all up for them. And then I think I ended up making them dinner afterwards and then just making sure that everything was set and good. That’s a particular way that I lend my help in the poly-fam. I’m a ‘fixit’ and a strong person, and that gets to be my area.*

Many participants discussed a willingness amongst all chosen family members to “*throw out a little extra money for people*,” especially those who make “*more money than most of the queer kids we know*,” as Grey said. About a couple members of his chosen family who are gainfully employed, Grey said, “*they are often the ones that not only I but other people in the community lean on if they need something.*” These exchanges are rarely kept as a tally and are often repaid through alternative means, like providing skills, time, and care. “*Our relationships aren’t zero-sum transactions*,” Grey concluded. Andres echoed this sentiment: “*As somebody in the queer community a lot of us end up having to rely on these very interconnected, interdependent support systems of close friends*.” Lydia agreed, indicating that while her chosen family members are a close group of people, she recognizes a sense of family that pervades the queer community at large: “*I think* [chosen family] *is just really important because a lot of times, especially the queer community*—[we don’t] *get that kind of support a lot, and so we have to make our own family*”.

Participants discussed mutual aid practices of sharing property and housing, financial resources, skills-based, and other material resources within their chosen family networks specifically. Additionally, participants referenced an implicit orientation to the principles of mutual aid as a pervasive practice throughout the queer communities in which they participate.

## 4. Discussion

In this study, we used unanswered questions about the theoretical and practical relationship between *care* and *family* as a launching point for exploring the ways in which Q/T individuals and communities navigate, negotiate, and challenge the existing bio-legal definitions of family in terms of access to formal institutions of health care; as well as engagement in informal mutual aid. By conducting original research on the operationalization of “chosen family,” we have taken up Newman’s [44] call to value and understand queer families not only for what they reveal about living with difference, but also for how they help us rethink what we assume to be normal. Few-Demo et al. ([45], p. 77) urge a paradigm shift wherein scholars “*ask different questions about all families*” (emphasis original). Our findings work to contribute to this call to action. 

For a study whose primary objective is to understand how Q/T chosen family members navigate health, illness, and the mutual provision of care, the authors were surprised by how scarcely populated that data were with conversations about care giving and care receiving *within* medical settings are. As described by participants, the locus of “care” amongst chosen family members is largely based outside of and beyond formal medical institutions. Rather, by default, care originates from informal, self-created and self-maintained networks. The tendency of this theme is due in part to the quality of our interview protocol, which was oriented toward understanding the theoretical links between *chosen family* and *care* at large, using health care and medical settings as a proxy for care giving in general. The intention to use care as a metric for articulating chosen family influenced this quality of the data. Nevertheless, despite the frame of our protocol, participants did not speak about experiences within medical settings unless prompted to do so, and even when probed they spoke only topically about doctor’s visits and formal medical procedures. This suggests that chosen family is constituted through caring relationships that largely operate outside of—or in contraindication to—medical settings or institutional contexts. This finding indicates that an ethic of care amongst chosen family is incongruent, or even illegible, within the context of medical settings. The tension between formal medical settings and chosen family as a primary site of care emerged as an unanticipated finding. However, given that care as a function of chosen family is not easily tracked, legitimated, or made intelligible by social institutions and other regulatory schemes, this finding is logical [21]. Therefore, further studies that expressly investigate chosen family experience *within* medical settings—rather than merely using health care as a metric for understanding chosen family as a function of care at large—is needed in order to more deeply and intricately understand chosen family experiences with respect to providers, health care settings, health care policies, medical practices, and their sequalae.

This information may be useful to mental health and other human services professionals in understanding the deep role chosen family plays within Q/T individuals’ lives. The ability to use this framework to discuss nuanced support structures and systems can be an effective mechanism to help those who may be disconnected from family of origin and who may require guidance to understand the ways in which they can create community and family who are supportive of their health and well-being. These findings should also be considered by health care workers and human service professionals who may tend to define care as solely connected to traditional medical settings. With this knowledge, home care and support plans can be tailored to include multiple people upon which a patient may rely, or who may rely upon them. Additionally, recognizing the important role chosen family plays in advocacy, care support, and emergency support may be useful in educating health care professionals about refraining from making assumptions, and instead asking more open-ended questions when it comes to discussing how an individual is cared for at home, and who may be attending appointments and procedures as an advocate.

Limitations to our study may be improved upon in future investigations with this population. On our demographic screening, we used ratio level data collection to ascertain age, and income level. We utilized language derived from the 2010 US Census [46] for nominal race/ethnicity categories. We offered a combination of nominal select-all-that-apply and self-report options for gender identity and sexual orientation, and disabilit(ies) and/or illnesses. Despite our efforts to be consistent and accurate, the ratio level data were not useful for the purposes of our study, nor were the Census-derived nominal race/ethnicity categories. With a sample size of (*n* = 11), nominal and ratio level categories were not necessary and in fact bore less accurate, cruder information than if participants had self-report options for each of the demographic items. We did not offer a self-report or select-all-that-apply options for pronouns, but rather instructed participants to write their pronouns at the top of the screening sheet as well as select a pseudonym, if preferred. This was a faulty practice, because on occasion, the interviewer forgot to ask for pronouns, resulting in a lack of information. In those cases, we refer to participants with “they/them” pronouns. 

Additionally, our sample was limited in its demographic diversity. Our sample was comprised mostly of young adults under the age of thirty. Few had children or cared for older adults, what Weston refers to as *“generational depth”* ([2], p. 180). Therefore, the quality of care giving as well as the legal negotiations involved in generational depth were not raised. Our study was also limited with respect to racial diversity. Our sample included few participants of color, though none identified as Black. This is significant because two participants, who are both white, raised awareness that their Black friends and friends of color relate with “chosen family” using principles of mutual aid yet without naming these communities or practices as such; and without such formulations of family existing as auxiliary to bio-logical family, or being otherwise unusual. In this sense, “chosen family” signifies the queering of middle and upper middle-class white cultures. Further research that critically engages the racial and socioeconomic conditions that produce chosen family, with particular focus on the ways in which intersections of racial and socioeconomic difference obligate chosen families to negotiate formal medical systems and mutual provision of care for health and illness, is needed. 

Finally, while the information gathered in this study was incredibly rich, it did not focus on the experience of chosen family specifically within medical contexts for the most part, as originally conceptualized by the authors. Further studies may wish to reframe how investigators ask about care and health in order to garner this information. As many Q/T people identify non bio-legal individuals as their emergency contacts, POA, and other legal custodians, and use them for support in these contexts, there is a need for understanding how affirming—or not—health care professionals are of the roles chosen family play in medical settings, and how comfortable Q/T people are of naming their chosen family as such.

## 5. Conclusions

This study explored some of the nuance in how Q/T individuals operationalize chosen family regarding health, care, and well-being. Several themes emerged, some focusing on the definitions and conceptualization of chosen family, and three additional themes specifically regarding navigating medical systems, leaning on each other, and mutual aid, were more deeply explored. Participants focused more on the concept of care and well-being support from their chosen family members, including providing advocacy in medical contexts, asymmetrical support and use of one another as emergency contacts and POA; organizing around health needs, assisting one another through changes in relationship contexts; sharing material resources and providing no-strings-attached support of one another. The results of this investigation can be useful for human service professionals and health care workers alike in better understanding the support that chosen family can offer Q/T individuals, and the kinds of care responsibilities that Q/T individuals hold for their chosen family members. Findings indicate the pressing need for health care and human service professionals to recognize and legitimize the non-traditional systems of care that exist outside extant medical settings. Clearer understanding of “chosen family” may enable health care professionals to improve upon service delivery to Q/T individuals; upon the ways in which formal medical institutions are amenable to and inclusive of chosen family formulations. 

## Figures and Tables

**Table 1 ijerph-17-07346-t001:** Sample Demographics.

Demographic Category (*n* = 11)	Responses	*n*	%
Gender Identity			
Woman		4	36.30%
Woman AND Nonbinary		1	9.10%
Transmasculine		1	9.10%
Nonbinary		2	18.20%
Transmasculine AND Nonbinary		1	9.10%
Transmasculine AND Nonbinary AND Genderqueer		1	9.10%
Transmasculine AND Agender		1	9.10%
Sexual Orientation/Identity			
Bisexual		5	45.50%
Pansexual		1	9.10%
Bisexual AND Queer		1	9.10%
Bisexual AND Queer AND Pansexual		1	9.10%
Pansexual AND Asexual		1	9.10%
Pansexual AND Demisexual		1	9.10%
Pansexual AND Queer		1	9.10%
Age Range			
18–25		3	27.30%
26–35		7	63.60%
36–45		1	9.10%
Race/Ethnicity			
Asian and/or Pacific Islander, Filipino, Chinese		2	18.20%
Asian and/or Pacific Islander AND White		1	9.10%
Middle Eastern, North African, and/or Chaldean AND White		1	9.10%
American Indian, Native Alaskan and/or Native Hawaiian AND White		1	9.10%
White		6	54.50%
**Income**			
Below $20,000		5	45.50%
$20,001-$30,000		2	18.20%
$40,001-$60,000		3	27.30%
$60,001-$80,000		1	9.10%
Highest Level of Education			
General Educational Development		1	9.10%
High School diploma		1	9.10%
Bachelor’s degree		2	18.20%
Master’s degree		4	36.30%
Doctorate degree or Professional degree		3	27.30%
Disability(ties) and/or illnesses			
Chronic pain, mental health, hard of hearing		1	9.10%
Depression/anxiety, post-traumatic stress disorder		1	9.10%
Mental health disability		1	9.10%
Post-traumatic stress disorder, generalized anxiety disorder, premenstrual dysphoria disorder, depression		1	9.10%
Severe mental illness, physical impairments		1	9.10%
Autism spectrum, chronic illness, lupus		1	9.10%
Depression		1	9.10%
Epilepsy, autism, chronic anxiety		1	9.10%
Mad/neurodiverse		1	9.10%
Obsessive compulsive disorder, anxiety, depression		1	9.10%
N/A		1	9.10%

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
