# Peer review of "“We Just Take Care of Each Other”: Navigating ‘Chosen Family’ in the Context of Health, Illness, and the Mutual Provision of Care amongst Queer and Transgender Young Adults"

_ijerph, 2020, doi:10.3390/ijerph17197346_

Round 1

Reviewer 1 Report

The aim of this paper was to explore the relationships between LGBTIQ+ people, chosen family, and modes of care. It furthers understandings of kinship, particularly within the LGBTIQ+ community where normative biological models of kinship may not necessarily apply. While aiming to focus primarily on chosen family in the context on medical support and care, the findings revealed a more general understanding of how kinship can function in the LGBTIQ+ community. 

The strength of this paper is in its contribution to kinship studies, particularly as kinship within LGBTIQ+ communities is under-researched. It serves to shift how we understand family beyond biological and legal structures to that of chosen families, and provides insights into how these chosen family can function. I would have perhaps appreciated slightly more discussion about the non-medical specific findings, but as the paper intended to focus on how kinship functions in medical/healthcare settings, I appreciate that such a discussion may not have been possible within the constraints of this particular paper. Is it possible to add in at least a sentence or two about the what the authors think the implications are of the non-medical findings? I found these findings particularly interesting and would love to know more what the authors thought about them. Additionally I think the paper would benefit from greater consistency around the usage of Q/T, LGBTQA, and LGBTQ acronyms - I have no preference to which one is used but there are a few different ones and no explanation of how queer is used in this context (i.e. is it used as an umbrella term for people of diverse sexual orientations, or is it used as a specific sexual orientation. Both are fine but some clarification would be useful).

Specific comments:

Line 54: This paragraph starts abruptly with the move to discuss the Bay Area - can you reword/soften the transition.

Line 60: Start of the sentence is clunky, please reword e.g. "for the purpose of this study..."

Line 92: Acronym is different and there's not definition of how queer is used.

Line 122: Same as above.

Line 126/section 2 more broadly: Can you please mention in the methods how many interviews were conducted just for clarity. 

Line 192: Sentence is slightly clunky/confusing, can you add a comma and the word "and" after demisexual.

575-8: Does this quote need to be indented to match others of similar lengths?

Author Response

Reviewer 1:

  1. The strength of this paper is in its contribution to kinship studies, particularly as kinship within LGBTIQ+ communities is under-researched. It serves to shift how we understand family beyond biological and legal structures to that of chosen families, and provides insights into how these chosen family can function. I would have perhaps appreciated slightly more discussion about the non-medical specific findings, but as the paper intended to focus on how kinship functions in medical/healthcare settings, I appreciate that such a discussion may not have been possible within the constraints of this particular paper. Is it possible to add in at least a sentence or two about the what the authors think the implications are of the non-medical findings? I found these findings particularly interesting and would love to know more what the authors thought about them. Additionally I think the paper would benefit from greater consistency around the usage of Q/T, LGBTQA, and LGBTQ acronyms - I have no preference to which one is used but there are a few different ones and no explanation of how queer is used in this context (i.e. is it used as an umbrella term for people of diverse sexual orientations, or is it used as a specific sexual orientation. Both are fine but some clarification would be useful).
    1. Thank you for raising this point. We have worked to more clearly articulate that we are exploring the notion of care as a constitutive factor in chosen family relationships as opposed to better understanding medical care giving outside of formal medical systems.
    2. Thank you for indicating these inconsistencies. We have changed all in-text acronyms to Q/T (queer and trans). We retain LGBTQIA2S+ when describing our recruitment materials for accuracy, and in one in vivo quote (Line 616). We added a sentence on Lines 219-221 to explain this use throughout out paper.
  2. Line 54: This paragraph starts abruptly with the move to discuss the Bay Area - can you reword/soften the transition.
    1. We have reworded/softened the transition between paragraphs using the following updated language: “Weston’s empirical on work on families of choice takes place against the backdrop of the HIV/AIDS crisis in San Francisco in the late 1980s. The ethnographic context of her findings is inextricable linked to her theoretical contributions: that families of choice and an ethics of care in the context of health and illness intersect both practically and theoretically is not incidental.”
  3. Line 60: Start of the sentence is clunky, please reword e.g. "for the purpose of this study..."
    1. Sentence reworded according to reviewer’s suggestion (Line 63)
  4. Line 92: Acronym is different and there's not definition of how queer is used.
    1. Updated to be consistent throughout paper (i.e. Q/T) and we have provided a rationale for our use of the term queer on Lines 219-221.
  5. Line 122: Same as above.
    1. Updated to be consistent throughout paper (i.e. Q/T)
  6. Line 126/section 2 more broadly: Can you please mention in the methods how many interviews were conducted just for clarity. 
    1. Sample size (N=11) added in methods (Line 207).
  7. Line 192: Sentence is slightly clunky/confusing, can you add a comma and the word "and" after demisexual.
    1. Grammatical/syntaxial errors have been corrected according to reviewer’s comments (Line 304).
  8. 575-8: Does this quote need to be indented to match others of similar lengths?
    1. We have indented this quote to match other quotes of similar length (Lines 711-713).

Reviewer 2 Report

This paper makes a meaningful contribution to the literature on chosen families and networks of care and support among queer and trans (Q/T) young adults. Findings of the paper highlight the volume of care work that is happening outside traditional medical settings. This holds implications not only for medical providers, but also for others who work with Q/T young adults. For example, colleges and universities might do more to recognize chosen family responsibilities among students, as these are unlikely to show up in studies of work, school, and family balance (and subsequent policies and practices) that rely on bio-legal definitions of family.

The paper also contributes to an emerging body of work on the lives and relationships of queer and trans people with disabilities and chronic illnesses.

I hope that my comments are supportive and helpful to the authors in developing this analysis.

  1. The article identifies Kath Weston’s (1991) Families We Choose as the last major work in this area. This elides important scholarship that does not necessarily use the term “chosen family” but advances the theory. I’m thinking, for example, of Marlon Bailey’s ethnography on ball communities in Detroit doing care work in kin terms, and his call for a shift in public health from studying risk to studying communities of support; Carla Pfeffer’s research on cisgender women partners of trans men doing transition-related medical/health care within and outside of the context of managed care, and the role of gender in shaping these dynamics; Sally Hines’ chapters on kinship and friendship, and on care networks, in her study of transgender identity, intimacy, and care; and Hil Malatino’s recent Trans Care <https://manifold.umn.edu/projects/trans-care>, which talks about trans care “within and against the medical-industrial complex.” Consider deepening the literature review to some extent, or at least removing the claim that substantial work on families of choice has not occurred. The value of the paper under review does not depend on the absence of this conversation – it adds something important to a conversation already under way.
  2. Kath Weston does acknowledge that bio-legal family may intersect with or, for some, preclude friends becoming kin, particularly for informants who experienced kinship as connected to racial or ethnic identity. Given that race and ethnicity are important to the conceptualization of family and chosen family, the paper should address the whiteness of the sample in more than the conclusion.

  3. Literatures on care networks within immigrant communities, Black communities, and transnational families have also emphasized support that exceeds bio-legal bonds, with terms such as othermothers and fictive kin emerging from Black studies. While the paper does not need to incorporate these bodies of work, it should also refrain from suggesting that families of choice are exclusively or originally queer.

  4. Consider making neurodiversity more prominent in how the paper is framed. Chosen family advocates, eating together, and other major themes are tied by the informants to neurodiversity, disability, and chronic illness. These seem as important for informants as sexual and gender identity. The spotlight on this intersection is a strength of the research. (Lines 472-480 start to get at this.)

Methods

  1. I recommend changing the methods section from passive to active voice. I recognize that this may be disciplinary preference. However, I think ownership of the research process and standpoint theorization are more in tune with the spirit of critical methodology that I sense in this work. The identities of the researchers matter particularly as the sample appears to have come though the researchers’ social networks.

  2. Tell us about the informants! I had to navigate to the tables at the end for this information. Include it in the methods section, before moving on to data analysis. Depending on the audience the authors hope to reach, it may be helpful to address somewhere the value of such a small sample.

  3. The demographics raise a question of class. 9 out of 11 informants have at least a college degree, and more than half have a graduate degree. This may simply be a function of recruitment. It may also indicate that the term “chosen family” is classed to some degree, even if the practice is not. (And if that’s the case, is how we talk about this creating the illusion of a more middle class phenomenon?) These are not issues or questions the paper needs to address in depth. It is already doing a great deal within the constraints of word count. But I do think acknowledging class, as well as race, is crucial for a careful analysis of these data.

Minor suggestions

  1. Lines 82-83 use “LGBTQA” when LGBTQ is used throughout the rest of the paper. Is there a reason for including asexual here/here only?

  2. Lines 99-100: The transition from problems with practitioners to the goals of the study does not feel intuitive to me as a reader. Consider creating a stronger bridge between these two paragraphs.

  3. Lines 262-263: “Responses to this question varied, with some participants including partners, some including bio-legal family members, and some including chosen family members.” Are partners not chosen family members? Until this point, I had included partners as “chosen family” in my mind; perhaps this has changed since legal bonds became possible for some? Either way, clarify this since the definition of chosen family is important to the heart of the paper.

I found this paper thought provoking and wish the authors the best in their revisions.

Author Response

Reviewer 2:

  1. The article identifies Kath Weston’s (1991) Families We Choose as the last major work in this area. This elides important scholarship that does not necessarily use the term “chosen family” but advances the theory. I’m thinking, for example, of Marlon Bailey’s ethnography on ball communities in Detroit doing care work in kin terms, and his call for a shift in public health from studying risk to studying communities of support; Carla Pfeffer’s research on cisgender women partners of trans men doing transition-related medical/health care within and outside of the context of managed care, and the role of gender in shaping these dynamics; Sally Hines’ chapters on kinship and friendship, and on care networks, in her study of transgender identity, intimacy, and care; and Hil Malatino’s recent Trans Care<https://manifold.umn.edu/projects/trans-care>, which talks about trans care “within and against the medical-industrial complex.” Consider deepening the literature review to some extent, or at least removing the claim that substantial work on families of choice has not occurred. The value of the paper under review does not depend on the absence of this conversation – it adds something important to a conversation already under way.
    1. Thank you for providing these resources and indicating the need to be more expansive in the literature review. We have removed the claim that substantial work on families of choice has not occurred and expanded the discussion on the existing literature (Lines 72-98).

  1. Kath Weston does acknowledge that bio-legal family may intersect with or, for some, preclude friends becoming kin, particularly for informants who experienced kinship as connected to racial or ethnic identity. Given that race and ethnicity are important to the conceptualization of family and chosen family, the paper should address the whiteness of the sample in more than the conclusion.
    1. Thank you for indicating the need to expand upon this discussion. We have foregrounded the intersectionality by including further discussion of this topic in a newly constructed final paragraph of the introduction (Lines 77-96)

  2. Literatures on care networks within immigrant communities, Black communities, and transnational families have also emphasized support that exceeds bio-legal bonds, with terms such as othermothers and fictive kin emerging from Black studies. While the paper does not need to incorporate these bodies of work, it should also refrain from suggesting that families of choice are exclusively or originally queer.
    1. Thank you for indicating the need to expand upon this discussion. We have foregrounded the intersectionality by including further discussion of this topic in a newly constructed final paragraph of the introduction (Lines 77-96)

  3. Consider making neurodiversity more prominent in how the paper is framed. Chosen family advocates, eating together, and other major themes are tied by the informants to neurodiversity, disability, and chronic illness. These seem as important for informants as sexual and gender identity. The spotlight on this intersection is a strength of the research. (Lines 472-480 start to get at this.)
    1. Thank you for indicating the need to expand upon this discussion. We have foregrounded the intersection of disability, neurodivergence, and queer identity by including further discussion of this topic in the introduction (Lines 142-153)

Methods

  1. I recommend changing the methods section from passive to active voice. I recognize that this may be disciplinary preference. However, I think ownership of the research process and standpoint theorization are more in tune with the spirit of critical methodology that I sense in this work. The identities of the researchers matter particularly as the sample appears to have come though the researchers’ social networks.
    1. Thank you for identifying and encouraging this shift in tone. We have changed the methods section from passive to active voice throughout (Lines 155-260) as well as added a reflexive comment about researcher’s positionalities (Lines 156-167).

  2. Tell us about the informants! I had to navigate to the tables at the end for this information. Include it in the methods section, before moving on to data analysis. Depending on the audience the authors hope to reach, it may be helpful to address somewhere the value of such a small sample.
    1. Thank you for this suggestion. We have added ample description of the sample in our ‘Materials and Methods’ section (Lines 207-250-X)
    2. We have added a note about the value of a small sample size for an exploratory phenomenological study design (Lines 251-260)

  3. The demographics raise a question of class. 9 out of 11 informants have at least a college degree, and more than half have a graduate degree. This may simply be a function of recruitment. It may also indicate that the term “chosen family” is classed to some degree, even if the practice is not. (And if that’s the case, is how we talk about this creating the illusion of a more middle class phenomenon?) These are not issues or questions the paper needs to address in depth. It is already doing a great deal within the constraints of word count. But I do think acknowledging class, as well as race, is crucial for a careful analysis of these data.
    1. Thank you for raising this demographic detail. We have added discussion of the relationship between education level and class status/identity in our ‘Materials and Methods’ section (Lines 227-236)  

Minor suggestions

  1. Lines 82-83 use “LGBTQA” when LGBTQ is used throughout the rest of the paper. Is there a reason for including asexual here/here only?
    1. LGBTQ
    2. LGBTQIA2S+ (recruitment section only)
      1. Thank you for indicating these inconsistencies. We have changed all in-text acronyms to Q/T. We retain LGBTQIA2S+ when describing out recruitment materials for accuracy, and in one in vivo quote (Line 616).  We added a sentence on Lines 219-221 to explain this use throughout out paper.

  2. Lines 99-100: The transition from problems with practitioners to the goals of the study does not feel intuitive to me as a reader. Consider creating a stronger bridge between these two paragraphs.
    1. Thank you for identifying this rough transition. We have added language that ties our study goals to the relationship to care more clearly (Lines 130-136).

  3. Lines 262-263: “Responses to this question varied, with some participants including partners, some including bio-legal family members, and some including chosen family members.” Are partners not chosen family members? Until this point, I had included partners as “chosen family” in my mind; perhaps this has changed since legal bonds became possible for some? Either way, clarify this since the definition of chosen family is important to the heart of the paper.
    1. Thank you for indicating this lack of clarity. We have added clarity about this question with respect the changing and contingent nature of access to legal marriage for same-sex partners (Lines 104-106).

Reviewer 3 Report

line 73, doesn't this reference have an exact date, volume number, and page numbers since it was accepted back in 2018?

line 169  I wish the authors would describe what "neurodiverse" means; is it a new term for mental illness or what?

line 213   I would like more discussion of using the term "identified with having severe mental illness, etc."   I thought that one was diagnosed by medical professionals with a mental illness or is this some type of self-diagnosis?  I wish there was more clarity about the meanings here.

line 221   the logic of this line wasn't clear to me.  If the doctors are experienced with doing breast reductions, why would they want to botch them up?

271  The English here is confusing.  "Chrissy is as bisexual"?  Did you mean Chrissy is a bisexual or Chrissy identifies as a bisexual?

271  "who does not identifies with"  should this be who does not identify?

277  It might be helpful to clarify/define framily when the term first comes up.

435  The English here is not clear.  "which is comprised of is non-monogamous" - is there an "a" missing here?

476  Here and in table 1 there may be a missing participant, n = 11 but data are only reported as if there were 10 participants, which is how the authors arrive at 90%.  The missing person should be explained.

Table 1.  What was the natal sex of the gender identities of strictly nonbinary?

Author Response

Reviewer 3:

  1. Line 73 doesn't this reference have an exact date, volume number, and page numbers since it was accepted back in 2018?
    1. Thank you for identifying this error in citation. We have updated citation 3 to accurately reflect the most up to date information.
  2. Line 169 I wish the authors would describe what "neurodiverse" means; is it a new term for mental illness or what?
    1. We have added clarification about this point in our newly written description of the sample (Lines 247-250).
  3. Line 213 I would like more discussion of using the term "identified with having severe mental illness, etc."   I thought that one was diagnosed by medical professionals with a mental illness or is this some type of self-diagnosis?  I wish there was more clarity about the meanings here.
    1. We have added clarification about this point in our newly written description of the sample (Lines 244-250).
  4. Line 221 the logic of this line wasn't clear to me.  If the doctors are experienced with doing breast reductions, why would they want to botch them up?
    1. To clarify this point for other readings who may find it confusing, we have qualified the participants quotation with bracketed explanation (Lines 333-334).
  5. Line: 271 The English here is confusing.  "Chrissy is as bisexual"?  Did you mean Chrissy is a bisexual or Chrissy identifies as a bisexual?
    1. Thank you for pointing out this typo. We have corrected the error by changing the word from “as” to “a.” (Line 338)
  6. Line 271 "who does not identifies with" should this be who does not identify?
    1. Thank you for pointing out this typo. We have corrected the error by changing the word from “identifies” to “identify.” (Line 338)
  7. Line 277 It might be helpful to clarify/define framily when the term first comes up.
    1. We have added a definition of framily as “a combination of the words ‘friend’ and ‘family’” (Lines 389-391).
  8. Line 435 The English here is not clear.  "which is comprised of is non-monogamous" - is there an "a" missing here?
    1. Thank you for pointing out this typo. We have corrected this typo by replacing the word “is” with the article “a” to clarify our meaning (Line 554).
  9. Line 476 Here and in table 1 there may be a missing participant, n = 11 but data are only reported as if there were 10 participants, which is how the authors arrive at 90%.  The missing person should be explained.
    1. Thank you for identifying this error. We have added data about the missing participant in the Dis/abilities category in Table 1. We have also changed the category to “Disabily(ies) and/or illnesses” to more accurately reflect the way we operationalize the category in the study.
  10. Table 1.  What was the natal sex of the gender identities of strictly nonbinary?
    1. We thank you for your question. We have decided not to include this piece of information because disclosing sex-assigned-at-birth of nonbinary participants is not a tenet of affirming research.